# Fertilization reduces aphid population growth but does not alter competitive exclusion between specialist and generalist species

Ayako Nakatani[1], Fumika Shindoh[2], Tatsuya Saga[2,3]*

**1** Akashikita Senior High School, Akashi, Japan, **2** Graduate School of Human Development and Environment, Kobe University, Kobe, Japan, **3** Institute for Promotion of Higher Education, Kobe University, Kobe, Japan

* tatsuya.saga@people.kobe-u.ac.jp

## Abstract

Understanding how resource availability shapes herbivore competitive interactions is crucial for predicting pest dynamics under changing agricultural practices. Interspecific competition among herbivorous insects is mediated by host plant quality, yet mechanisms underlying competitive outcomes under varying nutrient conditions remain unclear. We investigated how fertilization with a liquid N–P–K (5:8:12.5) solution diluted to 0.2% (v/v) affects population dynamics and competitive interactions between the legume-specialist *Megoura crassicauda* and the generalist *Aphis craccivora* on pea shoots (*Pisum sativum*) under fertilized and unfertilized conditions. Aphid population dynamics were monitored for 30 days in single-species and mixed-species treatments, and plant survival and biomass changes were assessed as whole-plant indicators of host performance under herbivory. Under single-species conditions, fertilization reduced *M. crassicauda* population growth (peak population decreased from 77.2±28.6 to 50.7±25.2 individuals per plant), while effects on *A. craccivora* were more modest (from 187.1±48.0 to 141.6±43.0). Area-under-the-curve analysis and peak abundance differed between species and were generally lower under fertilization. In mixed cultures, *A. craccivora* consistently dominated; *M. crassicauda* was excluded by day 25 under both nutrient regimes. Plants without aphids all survived, whereas mortality occurred in every aphid treatment except *M. crassicauda* with fertilization, indicating shorter survival times in aphid treatments than in controls. These results show that nutrient enrichment unexpectedly depressed herbivore population growth and did not relax interspecific competition. Competitive hierarchy was stable across nutrient regimes, with the generalist excluding the specialist.

## Introduction

In recent decades, global environmental changes such as climate change, land-use shifts, and nutrient enrichment have been recognized as major drivers of ecological

**Data availability statement:** All underlying raw experimental data are deposited in Zenodo at: https://doi.org/10.5281/zenodo.17667683. Statistical analysis results are provided as Supporting Information files.

**Funding:** AN received funding from the JST Global Science Campus ROOT Program. The funders had no role in study design, data collection and analysis, decision to publish, or preparation of the manuscript.

**Competing interests:** The authors have declared that no competing interests exist.

processes. These changes can alter plant biomass, tissue structure, nutritional quality, and chemical defenses. As a result, herbivorous insects that depend on these plants are affected in various ways, including distribution, feeding behavior, growth, and reproductive success [1]. Moreover, these alterations in plant condition can cascade through insect communities, influencing interactions between co-occurring herbivores.

Of particular interest is the growing recognition that plant-mediated indirect interactions among herbivores may shape competitive dynamics. Prior herbivory by one species can reduce plant quality, negatively affecting the population growth of subsequently arriving species [2,3]. For instance, Staley et al. [4] demonstrated that a plant's nutrient status could shift the outcome of interspecific competition among aphid species. However, most studies have focused on sequential rather than simultaneous interactions, and the mechanisms by which nutrient availability alters competitive outcomes remain poorly understood. In other cases, intraspecific plant interactions can alter secondary metabolite concentrations, changing herbivore preferences and distributions [5]. Additionally, parasitic plants can influence competitive interactions among host plants and indirectly alter insect assemblages by modifying plant defense–competition trade-offs [6]. These findings underscore the importance of considering herbivore–plant–plant or herbivore–plant–herbivore triads, even in systems that appear to involve only two interacting species.

Understanding such interactions requires examining how resource quantity and quality shape the intensity and outcome of interspecific competition. Classical ecological theory predicts that interspecific competition leads to either competitive exclusion or coexistence through niche partitioning [7]. However, empirical studies show that coexistence is often facilitated by asymmetric resource use or environmental variability [8], and the conditions determining competitive outcomes under varying resource availability remain poorly understood. For example, two hermit crab species in Japan have been shown to coexist by selecting different shell types [9]. Similarly, Juliano [10] demonstrated that food quality alters the relative strength of intra- and interspecific competition between two Aedes mosquito species, ultimately determining which species dominates under resource-limited conditions. Yet despite these advances, we lack a mechanistic understanding of how plant nutrient status influences the competitive balance between specialist and generalist herbivores—a knowledge gap with important implications for predicting pest dynamics under changing agricultural practices.

In this study, we focus on aphids (Aphididae), phloem-feeding insects that are sensitive to both plant quality and population density [11,12]. Aphids reproduce rapidly through parthenogenesis and exhibit density-dependent wing dimorphism, producing winged morphs under crowding that disperse to new hosts. These traits make aphids an excellent model for studying population-level responses to environmental variation. Moreover, aphids are major agricultural pests worldwide, making competitive dynamics directly relevant to pest management.

Many studies show that aphid population growth responds to plant nutritional status and defensive chemistry, with positive effects under moderate enrichment but

variable outcomes depending on fertilizer regime, host traits and aphid diet breadth [13–16]. Many studies show that aphid performance responds to plant nutritional status. Low nutrient availability reduces fecundity and survival through enhanced secondary metabolites and reduced nitrogen content [17]. Conversely, fertilization may improve aphid population growth by promoting plant growth [18]. However, fertilization effects are context-dependent: high nutrient inputs can produce no net or even negative effects when they alter plant physiology or modify mutualisms and enemy pressure [19,20]. Moreover, top-down forces in the field often attenuate the strong bottom-up effects observed in laboratory assays [21]. Despite this complexity, most research has focused on single-species responses to plant quality [13–16], with limited attention to how nutrient conditions alter interspecific interactions. Moreover, plant nutrient status can mediate competition among herbivores, including aphids, with high nitrogen availability sometimes alleviating plant-mediated competition observed under low nutrients [4]. In addition, subordinate aphid species may behaviorally detect dominant competitors and respond by reducing reproductive output or dispersing to avoid costly interactions [22], while natural enemies, climatic context, and local densities further modulate competitive outcomes [1,19–21]. Yet, we still lack a clear mechanistic understanding of how host-plant nutrient status shifts the magnitude and direction of interspecific competition on shared hosts—particularly between specialist and generalist aphids. To address this gap, we experimentally investigated how plant nutrient levels affect the competitive dynamics between *Megoura crassicauda* and *Aphis craccivora*, two aphid species that commonly co-occur on leguminous plants. We reared *M. crassicauda* and *A. craccivora* on pea shoots (*Pisum sativum*) under two nutrient regimes (fertilized and unfertilized), either separately or together. This experimental design allows us to distinguish between direct effects of plant nutrition on individual species performance and indirect effects mediated through interspecific competition—a distinction critical for predicting community-level responses to environmental change.

Our primary aim was to determine whether plant nutrient enrichment promotes coexistence by alleviating competition, or conversely, intensifies competition by accelerating population growth and resource consumption. We tested two predictions based on resource competition theory [8,23,24] and plant quality effects [14,19]. Accordingly, we tested two predictions: (1) Under high-nutrient conditions, increased resource availability will relax interspecific competition, allowing both species to persist. This prediction is supported by evidence that high nitrogen availability can alleviate plant-mediated competition among herbivores [4]. (2) Under low nutrient conditions, limited resources will intensify competition, resulting in the exclusion of one species. We quantified plant response at the whole-plant level (survival time and biomass change) as integrative indicators of host plant response to herbivory, rather than physiological tolerance. Comparing single-species with mixed-species assays allowed us to separate direct fertilization effects on each species from effects mediated by interspecific competition.

## Materials and methods

### Study organisms

This study used two aphid species from the family Aphididae: the legume specialist *Megoura crassicauda* and the generalist cowpea aphid *Aphis craccivora* (Fig 1a, b). *M. crassicauda* is distinguished by its green body and dark antennae, cornicles, and cauda, and typically measures from 3.0 to 4.5 mm in length. It feeds primarily on leguminous plants, especially *Vicia faba*. *A. craccivora* has a black body and is slightly smaller (2.5 to 4.0 mm), with a wide host range within Fabaceae. In natural habitats, both species frequently co-occur on the same host plant. Both species were observed co-occurring on the same host plants during field collections for this study.

### Host plants

Commercially available pea shoots (*Pisum sativum*) were used as host plants. Approximately 60 individual shoots, each measuring 15–20 cm in height, were transplanted into plastic pots (6 cm diameter × 5.5 cm high) filled with vermiculite one week before the experiment. Plants were grown in a controlled indoor environment (20–25°C, natural light) and divided into three groups: 44 for experimental use, 10 for dry mass reference measurements, and 6 as reserve. To estimate

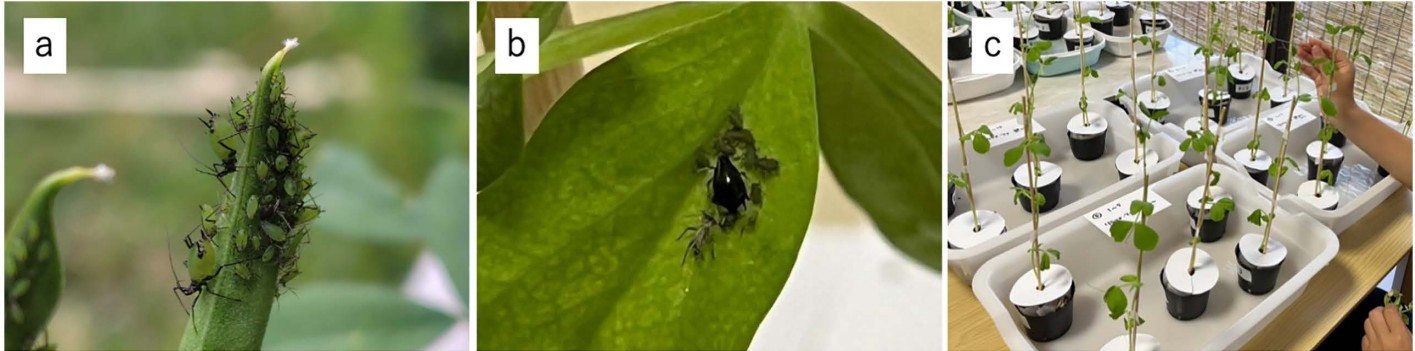

**Fig 1. a)** *Megoura crassicauda*, **b)** *Aphis craccivora*, **c) Laboratory experimental setup.** Each pot was continuously maintained in a container with 2 cm of standing water to prevent aphids from dispersing between pots.

pre-treatment biomass, 10 plants were oven-dried at 60°C for two weeks beginning three days before the experiment, and weighed using a precision balance (EK-i series, precision = 0.01 g).

## Aphid collection and inoculation

On April 29, 2024, approximately 300 wingless adults of each species were collected from a mixed-crop field in Tamatsu, Nishi-ku, Kobe, Japan (34.7°N, 135.0°E). We selected aphids based on visual assessment of vigor (active movement, intact appendages, normal coloration) and uniform body size (adult individuals of similar length within species-specific ranges). The same day, the experimental treatments were initiated by introducing aphids directly onto the host plants under laboratory conditions. Species identification followed the morphological keys in Moritsu [25], a standard reference for Aphididae taxonomy in Japan.

## Experimental Design

Six treatment groups (A-F) and two control groups (G and H) were established, as summarized in Table 1. Each plant was placed in a shallow container filled with either water or a diluted fertilizer solution (see below), and positioned at least 10 cm apart to prevent cross-contamination. To prevent aphid dispersal between treatments, each pot sat in an individual plastic tray (24 × 17 × 5 cm) containing a 2 cm water layer that acted as a physical moat. Together with wing removal upon alate detection, this setup effectively isolated each plant (Fig 1c). In mixed-species treatments, two individuals of each

**Table 1. Summary of experimental conditions.**

| Group | Fertilizer | Aphid species | No. of replicates | No. of aphids per plant |
|---|---|---|---|---|
| A | No | *M. crassicauda* only | 6 | 4 (wingless adults) |
| B | No | *A. craccivora* only | 6 | 4 (wingless adults) |
| C | Yes | *M. crassicauda* only | 6 | 4 (wingless adults) |
| D | Yes | *A. craccivora* only | 6 | 4 (wingless adults) |
| E | No | Mixed (2 + 2) | 10 | 2 of each species |
| F | Yes | Mixed (2 + 2) | 10 | 2 of each species |
| G | No | None (control) | 10 | 0 |
| H | Yes | None (control) | 10 | 0 |

aphid species were introduced to each plant. In single-species treatments, four individuals of the respective species were used. Control groups received no aphids and were used to assess plant survival and biomass in the absence of herbivory.

The fertilizer solution was prepared by diluting Hyponica liquid fertilizer (Kyowa Co., N–P–K ratio of 5:8:12.5) to 0.2% (i.e., 500-fold dilution) in water. This solution was provided to all fertilized treatments throughout the experiment. Our census schedule and treatment structure follow common practice in nutrient–aphid assays (e.g., [4]).

## Monitoring and measurements

The experiment lasted 30 days. Aphid populations were visually counted every two days using tally counters. Water or fertilizer solution was replenished every four days to maintain consistent soil moisture levels. When winged morphs appeared, we cut their wings using fine scissors to prevent plant-to-plant migration.

After 30 days, all plants were removed from their pots, gently cleaned of soil and aphids, blotted dry, and weighed to determine fresh biomass. Each plant was then oven-dried at 60°C for two weeks and weighed again to determine final dry biomass. The duration of plant survival (in days) was recorded for each plant. Plant response to herbivory was quantified at the whole-plant level using survival time and biomass change; physiological tolerance traits were not measured.

## Statistical analysis

All statistical analyses were conducted using R version 4.5.0 [26] with the following packages: tidyverse, survival, survminer, ggplot2, dplyr, car, emmeans, and multcomp. Statistical significance was set at $p < 0.05$. Three datasets were analyzed: (1) aphid population trajectories, (2) plant survival time, and (3) change in dry biomass (post- vs. pre-experiment).

To analyze aphid population growth, quadratic regression models ($y = ax^2 + bx + c$) were fitted to time-series data for each treatment group [27]. Treatment effects were evaluated using polynomial regression with interaction terms, and pairwise comparisons conducted using linear hypothesis testing from the car package [28].

To compare host plant survival times among treatment groups, Kaplan-Meier survival curves were constructed using the survival package, and differences were assessed using log-rank tests [29,30]. Plants surviving to day 30 were treated as censored observations. The Benjamini-Hochberg correction [31] was applied to adjust for multiple comparisons.

To test whether fertilization differentially affected the two aphid species, we analyzed single-species treatments (A–D) using two-way ANOVA. Species (*M. crassicauda*, *A. craccivora*) and Fertilizer (unfertilized, fertilized) were included as fixed factors with their interaction term. Response variables were log-transformed ($\log_e(1 + value)$) to meet assumptions of normality and homoscedasticity. We tested three summary measures per pot—AUC (area under the curve), peak abundance, and end-point count—representing cumulative, maximum, and final population size, respectively.

Changes in plant biomass were analyzed by subtracting pre-treatment dry mass from post-treatment dry mass. These differences were compared across treatments using ANOVA followed by Tukey's HSD test for multiple comparisons [27]. Significant interactions were interpreted as evidence for species-specific responses to fertilization.

## Ethics statement

According to the Regulations for Animal Experimentation and the Ethics Guidelines for Research Involving Human Subjects of Kobe University, this study did not require ethical approval because it involved only plants and insects. Field sampling was conducted on private land owned by the first author's family with permission.

## Results

We present the findings in three sections: (1) effects of fertilization on aphid population growth, (2) outcomes of interspecific competition, and (3) plant survival and biomass.

### Effects of host plant nutrient condition (fertilizer: yes/no)

**Population dynamics differed between fertilizer treatments.** Under single-species rearing conditions, maximum aphid population sizes were greater in unfertilized treatments compared to fertilized ones. *M. crassicauda* showed higher maximum populations in unfertilized conditions (77.2 ± 28.6) than fertilized (50.7 ± 25.2; A vs. C: F = 17.53, *p* < 0.001). Similarly, *A. craccivora* populations were higher without fertilizer (187.1 ± 48.0) than with fertilizer (141.6 ± 43.0; B vs. D: F = 4.00, *p* = 0.009). Similarly, cumulative mean population sizes over the experimental period were higher in the unfertilized conditions (*M. crassicauda*: unfertilized 578 ± 196 vs. fertilized 292 ± 141; *A. craccivora*: unfertilized 985 ± 230 vs. fertilized 726 ± 257). The mean daily number of aphids per plant under single-species rearing conditions is shown in Fig 2.

To isolate species-specific responses to fertilization from competitive effects, we analyzed single-species treatments (A–D) using a two-way ANOVA (Species × Fertilizer). Summary metrics (AUC, peak abundance, end-point count) were log-transformed ($\log_e[1 + \text{value}]$) prior to analysis. Both Species and Fertilizer had significant main effects on AUC (F(1,20) = 12.17, *p* = 0.002; F(1,20) = 7.28, *p* = 0.014) and peak abundance (F(1,20) = 27.26, *p* < 0.001; F(1,20) = 6.57, *p* = 0.019), whereas the interaction was not significant (*p* > 0.17). For end-point counts, the Species × Fertilizer interaction was significant (F(1,20) = 8.94, *p* = 0.007), indicating differential fertilizer effects between the two species. Pairwise contrasts showed fertilization consistently reduced *M. crassicauda* population growth (unfertilized > fertilized; AUC *p* = 0.0087; peak *p* = 0.012; end-point *p* < 0.001), while no clear effect was detected for *A. craccivora* (all *p* > 0.10). These patterns are visualized with raw-data boxplots (Fig 3).

Boxplots show log-transformed AUC ($\log_e(1+\text{AUC})$) for *M. crassicauda* (green) and *A. craccivora* (purple) under fertilized and unfertilized conditions. Fertilization reduced *M. crassicauda* population growth but not *A. craccivora*.

Polynomial regression confirmed these patterns: fertilization significantly altered *M. crassicauda* growth trajectories (linear *p* = 0.003; quadratic *p* = 0.008), whereas *A. craccivora* showed no significant change (*p* = 0.229; Fig 2; Table 2; S1 Table).

*M. crassicauda* exhibited a bimodal population pattern in both treatments, with peak on days 7 (36 individuals) and 15 (68) under unfertilized conditions, and on days 3 (22) and 17 (45) under fertilized conditions. The second generation appeared between days 3 and 5, coinciding with the divergence in population trajectories due to nutrient treatment.

For *A. craccivora*, no substantial difference in mean population was observed between treatments up to day 10 (unfertilized: 48.6 ± 30.3; fertilized: 47.8 ± 13.1). From days 13–15, populations increased rapidly, with fertilized plants temporarily supporting more individuals (108.3 ± 50.3) on day 14. However, after day 15, the unfertilized treatment again supported higher populations, a trend that continued during the population decline phase.

**Outcomes of interspecific competition.** In mixed-species treatments, *A. craccivora* consistently dominated, with *M. crassicauda* reaching zero abundance by day 25 in all replicates. Daily monitoring revealed the following temporal pattern (Fig 2).

In the early stage (days 1–9 without fertilizer; days 1–5 with fertilizer), *M. crassicauda* populations exceeded those of *A. craccivora* (e.g., unfertilized: 21.9 ± 12.0 vs. 7.9 ± 9.0 on day 5; fertilized: 25.3 ± 15.0 vs. 22.8 ± 11.6 on day 5). Subsequently, *A. craccivora* populations increased rapidly, surpassing *M. crassicauda* by day 11 in both fertilizer treatments (unfertilized: 28.6 ± 19.5 vs. 18.0 ± 9.7; fertilized: 56.1 ± 32.0 vs. 11.3 ± 8.9). After day 25, *M. crassicauda* was no longer observed on any plant (n = 20), whereas *A. craccivora* remained, despite a decline from its peak. Polynomial regression models were fitted to each treatment group (Table 2). One-way ANOVA revealed significant differences among mixed-species treatments (Groups 5–8) (F = 63.42, *p* < 0.001). In both fertilizer conditions, *A. craccivora* exhibited significantly steeper increases in population than *M. crassicauda* (*p* < 0.001 for both linear and quadratic terms, Tukey post hoc comparisons). Single versus mixed treatment comparisons:

- A vs. B (species comparison, unfertilized): F = 12.59, *p* < 0.001

- C vs. D (species comparison, fertilized): F = 31.27, *p* < 0.001

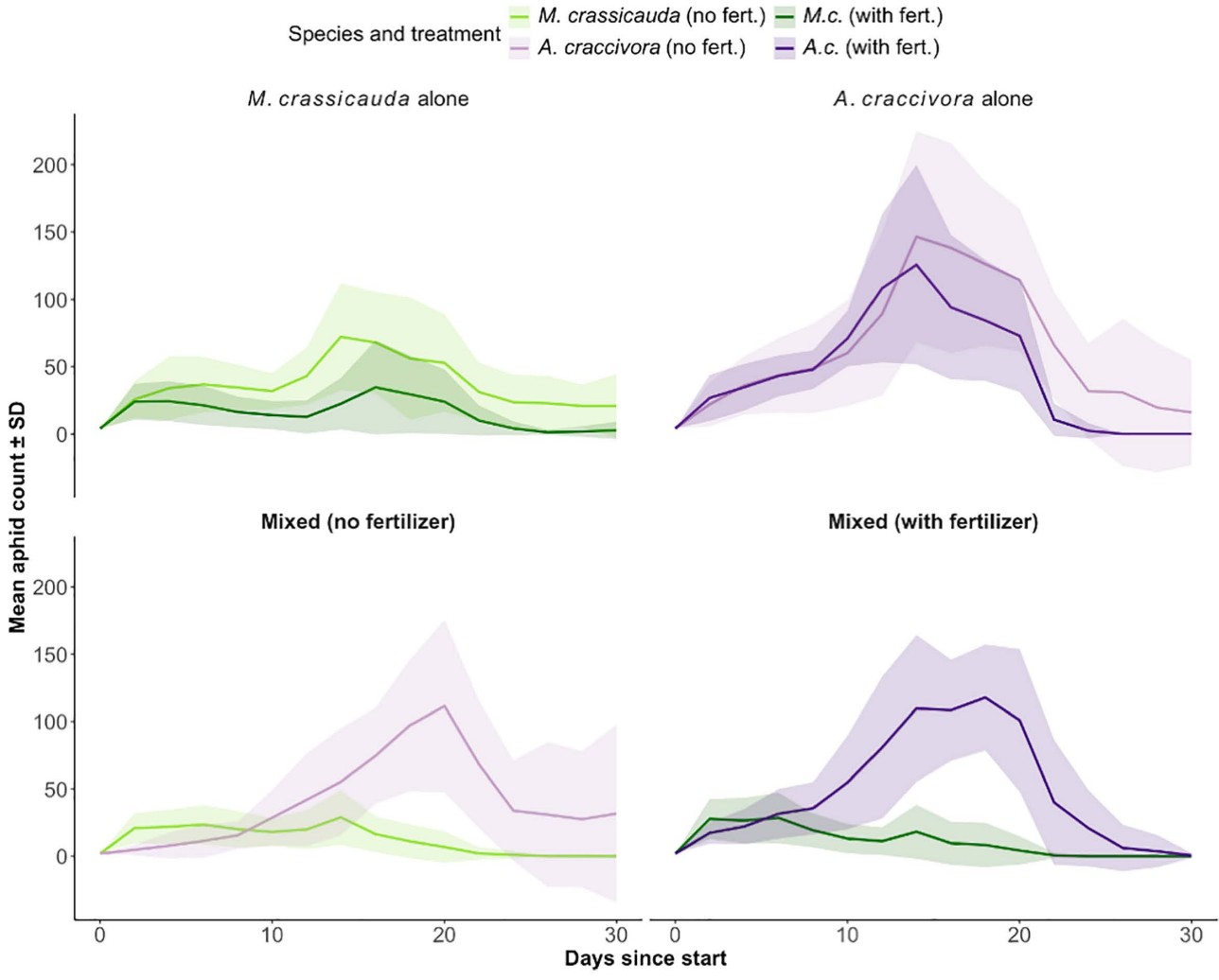

**Fig. 2. Aphid population dynamics under different treatments.** Each line shows the mean aphid count per plant (± standard deviation (SD)) over time. Panels separate fertilization (columns) and competition type (rows), and colors distinguish the two aphid species (*M. crassicauda* in green, *A. craccivora* in purple). This layout allows visual comparison of species responses across nutrient and competition treatments.

No aggressive interactions or direct exclusion were observed. Alate morphs emerged more frequently after day 17. Colonies of *M. crassicauda* tended to form on stems, whereas *A. craccivora* colonized the undersides of leaves.

**Host plant survival and condition.** Plant survival differed significantly among treatments. All plants in the control groups (G: unfertilized, no aphids; H: fertilized, no aphids) survived the full 30-day experimental period, exhibiting positive changes in biomass (G: +0.077 ± 0.083 g; H: +0.134 ± 0.089 g). In contrast, plant mortality occurred before day 30 in all aphid treatment groups except for Group C. In contrast, all aphid-infested groups showed negative biomass changes (g): A: −0.027 ± 0.048, B: −0.052 ± 0.023, C: −0.005 ± 0.026, D: −0.053 ± 0.015, E: −0.060 ± 0.032 g, and F: −0.105 ± 0.032.

Kaplan–Meier survival analysis revealed significantly shorter plant survival time in group D compared to group C (log-rank test, $p < 0.05$). Furthermore, when comparing control groups (G and H) with treatment groups (A–F), plant mortality occurred significantly earlier in the treatment groups ($\chi^2 = 17.2$, $p < 0.001$, Fig 4).

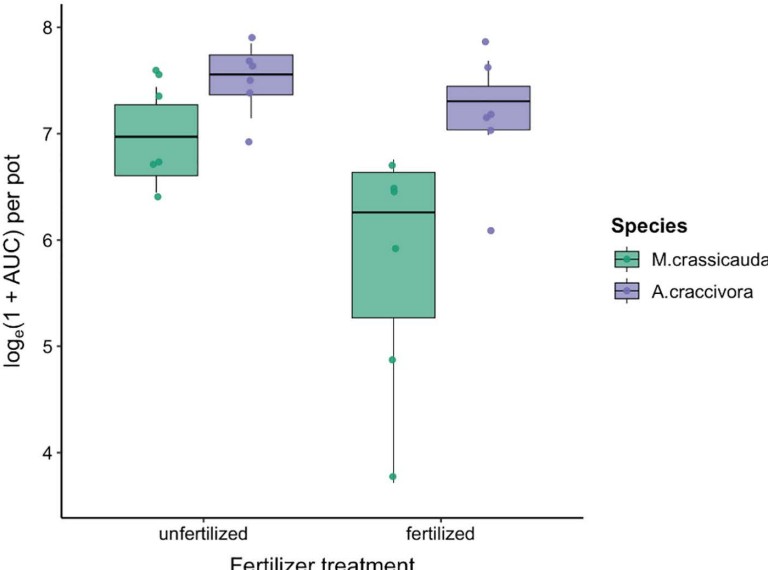

**Fig 3. Species-specific effects of fertilization on aphid population size.**

**Table 2. Summary of polynomial regression models for aphid population dynamics.**

| Group | Equation | R² | Adj. R² | F | p | σ |
|---|---|---|---|---|---|---|
| 1: *M. crassicauda* (unfert.) | y(t) = 36.156 + 4.328t - 142.459t² | 0.244 | 0.228 | 15.01 | < 0.001 | 26.02 |
| 2: *A. craccivora* (unfert.) | y(t) = 62.104 + 32.126t - 376.768t² | 0.380 | 0.367 | 28.50 | < 0.001 | 50.09 |
| 3: *M. crassicauda* (fert.) | y(t) = 15.448 - 43.118t - 59.273t² | 0.171 | 0.153 | 9.570 | < 0.001 | 16.75 |
| 4: *A. craccivora* (fert.) | y(t) = 45.385 - 100.396t - 324.010t² | 0.466 | 0.454 | 40.55 | < 0.001 | 37.67 |
| 5: *M. crassicauda* (mixed unfert.) | y(t) = 12.069 - 82.414t - 59.762t² | 0.318 | 0.310 | 36.66 | < 0.001 | 11.89 |
| 6: *A. craccivora* (mixed unfert.) | y(t) = 40.181 - 185.621t - 253.722t² | 0.257 | 0.248 | 27.15 | < 0.001 | 42.66 |
| 7: *M. crassicauda* (mixed fert.) | y(t) = 10.606 - 96.922t - 22.456t² | 0.265 | 0.255 | 28.25 | < 0.001 | 13.24 |
| 8: *A. craccivora* (mixed fert.) | y(t) = 47.013 - 8.249t - 444.857t² | 0.466 | 0.459 | 68.45 | < 0.001 | 38.03 |

Equations show the relationship between time (t, in days) and population size (y, number of individuals). All models were significant at $p < 0.001$.

Kaplan–Meier survival curves show the probability of plant survival over 30 days under different aphid infestation conditions. Groups A–F represent plants infested with either *M. crassicauda*, *A. craccivora*, or both, under fertilized or unfertilized conditions. Groups G and H are uninfested control groups. Significant differences in survival time were found between treatment and control groups (log-rank test, $\chi^2 = 17.2$, df = 1, $p < 0.001$). Among aphid treatments, only Group C (*M. crassicauda* with fertilizer) showed no plant mortality during the experimental period. Numbers in parentheses indicate sample sizes for each group.

## Discussion

### Effects of host plant nutrient conditions on aphid population growth

Time-series trajectories in Fig 2 indicate species-specific responses to fertilization: *M. crassicauda* shows altered growth trajectories (including an earlier divergence and bimodality), whereas *A. craccivora* changed little. We then corroborated these patterns with per-pot summaries and formal tests (Fig 3).

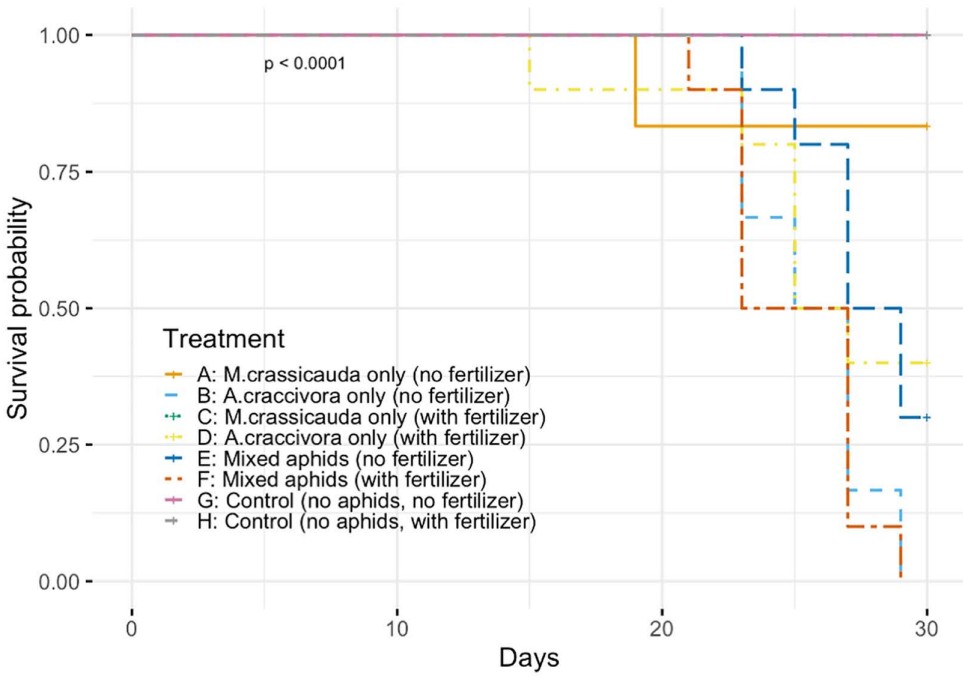

**Fig 4. Survival analysis of pea plants infested by different aphid treatments.**

Our attribution analysis using single-species treatments demonstrates a species-specific fertilizer effect: fertilization consistently reduced *M. crassicauda* across metrics, whereas *A. craccivora* was largely unaffected (Fig 3), indicating that nutrient effects depend on aphid identity.

This pattern suggests that fertilization altered host-plant quality in ways unfavorable to *M. crassicauda*. Plausible mechanisms include shifts in phloem amino acids and sugars, changes in leaf physical properties, and/or induced defenses; testing these alternatives will require plant chemical and physiological measurements. The contrast between the specialist (*M. crassicauda*) and the generalist (*A. craccivora*) is consistent with the idea that specialists can be more sensitive to specific host-quality changes, whereas generalists tolerate a wider range of chemistry [14,32,33].

## Outcomes of interspecific competition

The consistent dominance of *A. craccivora* over *M. crassicauda* in mixed treatments, regardless of fertilization status, demonstrates asymmetric competition leading to competitive exclusion. This outcome occurred despite *M. crassicauda* initially establishing higher population densities in the early phase of the experiment (days 1–9), suggesting that early colonization advantage does not guarantee competitive success. Similar patterns have been documented in ectomycorrhizal fungi, where species that gain initial competitive advantage through spore-based colonization can lose their dominance when competition shifts to mycelial-based interactions [34]. In some host-parasitoid systems, specialists with narrow host ranges are known to excel at host detection, although generalists may gain advantage under certain conditions.

The temporal dynamics revealed distinct phases of competition: initial establishment (days 1–9), competitive reversal (days 10–25), and exclusion (day 25 onwards). Such three-phase dynamics are consistent with theoretical models of competition–colonization tradeoffs, where early colonizers may be displaced by superior competitors depending on spatial resource structure and tradeoffs between dispersal and competitive ability [35,36]. The mechanisms underlying the competitive exclusion of *M. crassicauda* by *A. craccivora* remain to be determined. Although no direct aggressive

interactions were observed, several non-exclusive hypotheses could explain this pattern: (1) differential resource utilization efficiency, (2) chemical interference through honeydew or other secretions, or (3) differential responses to induced plant defenses. The spatial segregation observed (*M. crassicauda* on stems, *A. craccivora* on leaf undersides) suggests possible niche differentiation, yet this was insufficient to prevent competitive exclusion. Further research incorporating detailed behavioral observations and resource utilization measurements would help elucidate these mechanisms.

The differential impacts of the two aphid species on plant survival and growth may reflect their contrasting ecological strategies as specialist versus generalist herbivores. Generalist herbivorous insects often need to consume significantly more plant biomass when feeding on less suitable hosts in order to meet their nutritional requirements [37]. In contrast, specialist herbivores, having evolved tolerance to their host plant's defensive compounds, may achieve sufficient nutrition with less feeding and thus cause relatively lower plant damage [14]. Our findings align with recent meta-analyses showing that specialist herbivores generally cause less damage to their host plants compared to generalists [14], though the underlying mechanisms remain to be fully elucidated.

Consistent with this, in the present study, *M. crassicauda*, a specialist on fabaceous plants, caused relatively less plant mortality under fertilized conditions. In contrast, *A. craccivora*, a generalist, exerted consistently strong negative impacts on plant health irrespective of fertilization status. Differences in plant nutritional quality and defensive traits are known to significantly influence herbivore feeding strategies and population growth, particularly distinguishing specialists from generalists [38].

Furthermore, the observed fertilization effects may be linked to how plant nutritional status influences aphid feeding behaviors. Enhanced plant nutrition under fertilized conditions may have allowed *M. crassicauda* to meet its nutritional requirements more efficiently, though the specific mechanisms require further investigation. Conversely, *A. craccivora*, with broader dietary tolerance, likely maintained aggressive feeding regardless of plant nutritional status. This highlights how resource availability, plant tolerance, and plant resistance interactively influence herbivore-induced plant damage [39,40].

## Host plant responses and implications

Plant survival analysis revealed contrasting impacts of the two aphid species on host plant health. The complete survival of plants infested with *M. crassicauda* under fertilized conditions (Group C) contrasts sharply with the mortality observed in other aphid treatments (e.g., Group D: *A. craccivora* with fertilizer, mean survival = 23.3 ± 3.9 days), indicating species-specific differences in plant impact intensity. Specialist herbivores have typically evolved adaptations that enable them to effectively cope with specific host plant defenses [38], allowing them to obtain necessary nutrition with relatively less plant damage. In contrast, generalist herbivores, which feed on a wider range of plant families, often require increased plant biomass consumption when confronted with highly defended plant species, as their detoxification mechanisms are less adapted to specific defenses. Indeed, previous studies comparing specialist and generalist herbivore population growth on wild versus cultivated Brassica populations demonstrated that generalists caused greater plant damage than specialists due to their less specialized detoxification abilities [41].

The current study confirmed that *A. craccivora* consistently imposed greater physiological stress on host plants, evidenced by earlier plant mortality and more substantial biomass reduction. This pattern persisted across both single-species and mixed-species treatments, suggesting that *A. craccivora* depletes plant resources more rapidly and/or inflicts greater feeding damage compared to *M. crassicauda*.

The interaction between fertilization and aphid species effects on plant survival suggests complex tri-trophic relationships. In the literature, "tolerance" denotes recovery capacity after herbivory (e.g., [42]). While resource availability can modulate tolerance and resistance [39,40], we did not measure physiological or biochemical tolerance traits. Our survival and biomass results therefore indicate whole-plant response to herbivory, not tolerance per se. Under fertilized conditions,

plants infested with *M. crassicauda* (specialist) showed complete survival, whereas *A. craccivora* (generalist) caused mortality regardless of nutrient status, suggesting differential plant impacts that may involve tolerance mechanisms.

## Ecological and applied implications

These findings have important implications for understanding aphid community dynamics under changing agricultural management practices. The fertilization-mediated reduction in population growth of certain aphid species (specialists) suggests that particular nutrient management strategies might unintentionally offer pest control benefits. However, this effect is species-specific, and competitive displacement toward more damaging generalist species could lead to increased crop damage.

The consistent competitive dominance of *A. craccivora*, regardless of fertilization status, indicates that environmental modifications affecting resource availability may not alter fundamental competitive hierarchies between herbivore species. Consequently, predicting pest community composition under different management scenarios remains feasible within a certain range of environmental variation.

From an agricultural perspective, the observed differential impact of the two aphid species on plant health implies that competitive displacement favoring *A. craccivora* may result in increased crop damage even if overall aphid abundance remains constant. Previous studies support the hypothesis that generalist herbivores typically cause greater feeding damage to plants than specialist herbivores, reinforcing the importance of considering herbivore identity rather than simply abundance when making pest management decisions [14,38].

Fertilization may influence plant tolerance and resistance [39,40], but our study did not measure the underlying physiological traits required to test this mechanism. Thus, fertilization management may influence plant defensive capacity and damage outcomes, but mechanistic tolerance and resistance responses require targeted measurements before management recommendations can be drawn.

## Study limitations and future directions

Several limitations should be acknowledged. First, the experiment was conducted under controlled laboratory conditions using a single host plant species and a specific fertilizer composition. Field conditions involving multiple plant species, varying environmental stresses, and complex arthropod communities may yield different competitive outcomes. Second, the absence of natural enemies in our experimental system differs from field conditions where predation and parasitism can alter competitive outcomes. Third, the mechanisms underlying the observed competitive asymmetry remain unclear from our behavioral observations alone.

Our measurements were limited to aphid counts and plant biomass. These are population-level responses. Because our trait measurements were limited to population counts and biomass, we cannot identify the specific physiological mechanisms underlying the observed plant–aphid interactions. Future studies should measure a broader set of plant and aphid traits. For plants, these should include chlorophyll content, photosynthetic rate, stomatal conductance, phloem amino acids, and defensive metabolites. For aphids, we plan to measure reproduction rate, development time, adult longevity, and honeydew production. We will also include indicators of plant tolerance, such as growth rate, tissue allocation, and antioxidant activity. In addition, we plan to use metabolomic profiling, behavioral observations, and resource-use assays. Field experiments that include natural enemies will help confirm the results under realistic conditions.

These findings contribute to our understanding of how environmental changes may restructure herbivore communities and highlight the importance of considering competitive interactions in predicting ecological responses to anthropogenic change. The predictable nature of competitive outcomes demonstrated here provides a foundation for developing more effective pest management strategies under changing agricultural conditions.

Because our experiment was performed under controlled conditions without natural enemies or environmental heterogeneity, the direct extrapolation of quantitative results to field conditions should be cautious. Nevertheless, the observed

                                       

responses are consistent with field studies showing nutrient-driven shifts in aphid population growth [4,20], suggesting that our findings provide a useful mechanistic baseline for future field validation. This study represents a single 30-day laboratory trial. Results should be confirmed through temporal replication (independent experimental runs at different times) and greenhouse or field trials that include natural enemies.

## Supporting information

**S1 Table. Polynomial regression coefficients for aphid population dynamics models.**
(DOCX)

## Acknowledgments

The authors are also grateful to Akashi Kita High School for kindly providing laboratory equipment essential for the experiments. The authors are grateful to Dr. Mohamed Braham and an anonymous reviewer for their helpful comments on the manuscript.

## Author contributions

**Conceptualization:** Ayako Nakatani, Tatsuya Saga.

**Data curation:** Ayako Nakatani.

**Formal analysis:** Ayako Nakatani, Tatsuya Saga.

**Funding acquisition:** Ayako Nakatani.

**Investigation:** Ayako Nakatani.

**Methodology:** Ayako Nakatani, Fumika Shindoh, Tatsuya Saga.

**Project administration:** Tatsuya Saga.

**Supervision:** Tatsuya Saga.

**Visualization:** Ayako Nakatani, Tatsuya Saga.

**Writing – original draft:** Ayako Nakatani, Tatsuya Saga.

**Writing – review & editing:** Fumika Shindoh, Tatsuya Saga.

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
