## [Decision Letter · Decision Letter 0]

2 Oct 2025

Dear Dr.  Saga,

Thank you for submitting your manuscript to PLOS ONE. After careful consideration, we feel that it has merit but does not fully meet PLOS ONE’s publication criteria as it currently stands. Therefore, we invite you to submit a revised version of the manuscript that addresses the points raised during the review process.

Both reviewers concur that the study holds great merit and relevance, and the overall structure of the manuscript is satisfactory and generally understandable as well as relevant to a wide potential readership

The main issues that need addressing are, broadly, as follows:

Literature review in Introduction appears too sketchy and narrow for such a cutting edge topic in modern agronomy with increasing pressure of pest upsurge against crop production. For example, in Line 97-102, the authors state that “Many studies have examined how aphid performance is influenced by plant quality ….” And then go ahead to list only 2 such citations. There is a vast volume of recent literature on this subject from which the authors may draw, to help place their research into a suitable gap-niche (i.e. justification for the study)

The study’s aims are quite clear, and the hypotheses are very well articulated and backed with sound theoretical foundations that justify them, but these theoretical justifications need at least 2 relevant citations

Broaden the range of responses by both plants and aphids to perceived effects of fertilization (additional to counts and biomass measurements)

Explain clearly in response to Reviewer #1 to what extent your experimental protocol and results obtained may be practically applicable in real life/field situations outside the controlled laboratory condictions

The provided statement of experimental strategy for minimizing aphid movement inter-treatment (e.g. clipping wings) does not seem to suffice. Unless legs were also clipped, can’t the insects still disperse around freely?

Experimental method protocol needs much more detail, including reference to standard procedures, authority citations and instrumental/material detailing to support validation and repeatability/replication

Provide elaborate references/citations for the various statistical tools/equations/theories adopted for analyses

Provide at least one Figure showing a color-separated images of the study species of aphids

We look forward to receiving your revised manuscript.

Kind regards,

Nickson Erick Otieno, PhD

Academic Editor

PLOS ONE

Journal Requirements:

“AN received funding from the JST Global Science Campus ROOT Program.”

Additional Editor Comments:

Each of the reviewers has, in addition to the general comments and statements of concern or suggested changes, also offered additional specific comments in the article PDF which is attached to the authors in the submission platform. The authors must carefully access these, address each of the comments and, in resubmitting a thoroughly revised manuscript, include a separate document detailing how each of these issues has been addressed point by point, or a robust rebuttal as to why they disagree with suggested changes

The formatting of the document has not adhered to the guidelines of PLOS ONE. In assessing the revised manuscript, the strict compliance to formatting guidelines (structure, font sizes and styles as well as arrangement of the various component parts) will be the first criteria before the paper can progress any further, failure to which it will have to be rejected outright. For instance, in-text citation style uses numbers, not author-and-year; they must also be enclosed in square brackets, and listed in the corresponding number orders under References at the end.

Tables and their captions must be immediately below paragraphs they are first mentioned in;

Figure captions must be in the positions below paragraphs they are first mentioned in

Refer to submission guidelines for strict compliance will all requirements

Reviewers' comments:

Reviewer's Responses to Questions

**Comments to the Author**

1. Is the manuscript technically sound, and do the data support the conclusions?

Reviewer #1: No

Reviewer #2: Yes

2. Has the statistical analysis been performed appropriately and rigorously?

Reviewer #1: Yes

Reviewer #2: Yes

3. Have the authors made all data underlying the findings in their manuscript fully available?

Reviewer #1: Yes

Reviewer #2: Yes

4. Is the manuscript presented in an intelligible fashion and written in standard English?

Reviewer #1: No

Reviewer #2: Yes

Reviewer #1: Dear Authors,

Thank you for submitting your manuscript to Plus One. The study addresses a valuable question about how fertilization shapes aphid performance and interspecific competition on pea plants. While your work has merit, the manuscript, in its current format, does not meet the journal’s standards for publication. Several important methodological and interpretive issues need to be addressed, and I have outlined them below with guidance to help you prepare a stronger revision. A separate document with detailed, itemized suggestions is attached.

Main issues to consider and revise:

1- Trait scope and depth:

- The current trait set (aphid counts and plant biomass) is too limited to robustly infer mechanisms of plant response to aphid infestation and to establish the influence of fertilization on competitive outcomes.

- Consider adding additional plant responses (e.g., chlorophyll content, photosynthetic rate, stomatal conductance, tissue nutrient concentrations, defensive metabolites) and aphid performance metrics (e.g., reproduction rate, development time, adult longevity, honeydew production) to better capture trophic interactions and tolerance mechanisms.

2- Proof of fertilization effect on species-specific aphid numbers:

Clarify how fertilization differentially affected the two aphid species. Provide explicit experimental design details and statistical tests that support the claim that changes in aphid counts by species are attributable to fertilizer treatment rather than other factors.

3- Containment and movement between pots:

Describe how you prevented aphid movement between pots (physical separation, barriers, spatial layout, or any measures to minimize cross-transfer). If movement was possible, discuss its potential impact on results and how it was mitigated or accounted for in analyses.

4- Plant tolerance assessment

Biomass alone is insufficient to characterize plant tolerance to herbivory. Include additional indicators (growth rates, tissue allocation patterns, physiological stress indicators, biochemical markers) to support conclusions about tolerance under different nutrient regimes.

5- Repetition and clarity regarding trait choices

There are repeated points about the limited trait set. Consider consolidating these into a single, clear rationale for expanding trait measurements and present a concrete plan for the expanded suite of traits.

6- Methods: measurement references

The Materials and Methods section should include explicit references or protocols for all measurement techniques. Provide enough detail for replication, including instrument models, measurement conditions, sampling timing, and data processing steps.

7- Language and presentation

Consider a careful language edit to improve clarity and conciseness. Clear, precise phrasing will help readers understand the experimental design, results, and interpretation.

I hope you will consider a substantial revision.

Best regards

Reviewer #2: The paper is well written but it deals only with a laboratory experiment lasted only 30 days. I think replication is needed to confirm results. Also, Figure 1 is not very legible and difficult to interpret.

**Do you want your identity to be public for this peer review?** For information about this choice, including consent withdrawal, please see our Privacy Policy

Reviewer #1: No

Reviewer #2: **Yes: ** Pr. Mohamed BRAHAM

---

## [Author Response · Author response to Decision Letter 1]

9 Nov 2025

Dear Dr. Saga,

Thank you for submitting your manuscript to PLOS ONE. After careful consideration, we feel that it has merit but does not fully meet PLOS ONE’s publication criteria as it currently stands. Therefore, we invite you to submit a revised version of the manuscript that addresses the points raised during the review process.

Both reviewers concur that the study holds great merit and relevance, and the overall structure of the manuscript is satisfactory and generally understandable as well as relevant to a wide potential readership

The main issues that need addressing are, broadly, as follows:

Literature review in Introduction appears too sketchy and narrow for such a cutting edge topic in modern agronomy with increasing pressure of pest upsurge against crop production. For example, in Line 97-102, the authors state that “Many studies have examined how aphid performance is influenced by plant quality ….” And then go ahead to list only 2 such citations. There is a vast volume of recent literature on this subject from which the authors may draw, to help place their research into a suitable gap-niche (i.e. justification for the study)

L99-124: We have substantially expanded the Introduction’s literature review to reflect the breadth of work on nutrient–aphid interactions and competitive dynamics. We added recent syntheses and field studies (e.g., Abdala-Roberts 2019; Aguilera 2021; Lescano 2022; Åhman 2019; Hamann 2021) and clarified the study gap: how host-plant nutrient status changes the magnitude and direction of interspecific competition, especially between a specialist and a generalist aphid.

The study’s aims are quite clear, and the hypotheses are very well articulated and backed with sound theoretical foundations that justify them, but these theoretical justifications need at least 2 relevant citations

L133-144: We added theoretical justifications with relevant citations in the Introduction. Specifically, we now ground our hypotheses in modern coexistence and resource‐competition theory (Chesson 2000; HilleRisLambers et al. 2012; Barabás et al. 2018) and in nutrient-driven bottom-up effects in herbivore–plant systems (Ali & Agrawal 2012; Abdala-Roberts et al. 2019).

Broaden the range of responses by both plants and aphids to perceived effects of fertilization (additional to counts and biomass measurements)

- We agree that additional physiological or chemical measurements could deepen the interpretation of fertilization effects. However, our experiment was designed to focus on demographic outcomes (aphid abundance and plant biomass) which directly reflect resource competition strength.

- L496-505: We have added a note in the Discussion acknowledging this limitation and emphasizing that future studies should include physiological and biochemical traits to clarify the underlying mechanisms.

Explain clearly in response to Reviewer #1 to what extent your experimental protocol and results obtained may be practically applicable in real life/field situations outside the controlled laboratory condictions

- We agree that direct extrapolation to field conditions should be made cautiously. The experiment was conducted under controlled conditions to isolate nutrient and competition effects.

- L512-520: We now clarify in the Discussion that while absolute values may differ in the field, the direction of responses we observed aligns with field studies (e.g., Staley et al., 2011; Aguilera et al., 2021) and provides a theoretical basis for future field validations.

The provided statement of experimental strategy for minimizing aphid movement inter-treatment (e.g. clipping wings) does not seem to suffice. Unless legs were also clipped, can’t the insects still disperse around freely?

- We apologize for the insufficient description. In addition to removing wings from alate individuals, all pots were placed inside separate plastic trays filled with a thin layer of water to prevent aphids from moving between pots. This water barrier effectively restricted movement while avoiding any harm to the insects.

- L185-190: We have clarified this procedure in the revised Methods section.

Experimental method protocol needs much more detail, including reference to standard procedures, authority citations and instrumental/material detailing to support validation and repeatability/replication

- We appreciate this suggestion.

- L181-182, 197-199: We revised Materials and Methods to add procedural detail and authoritative citations. We now cite Moritsu (1983) for species identification and Staley et al. (2011) for experimental setup and census schedule. These changes improve transparency and reproducibility

Provide elaborate references/citations for the various statistical tools/equations/theories adopted for analyses

- L213-237: We have added authoritative citations supporting each statistical method, including polynomial regression and ANOVA/Tukey (Zar, 2014; Fox & Weisberg, 2019), survival analysis (Kaplan & Meier, 1958; Mantel, 1966), and multiple-comparison control (Benjamini & Hochberg, 1995). These additions ensure clarity, replicability, and methodological validation.

Provide at least one Figure showing a color-separated images of the study species of aphids

We have added images of both aphid species to Figure 1 (panels a and b).

Nickson Erick Otieno, PhD

Academic Editor

PLOS ONE

Journal Requirements:

We have revised the manuscript to comply with PLOS ONE's style requirements.

Thank you for pointing this out. The research was partially supported by internal funds from Kobe University that were provided through a JST (Japan Science and Technology Agency) program to support high school research collaboration. This support does not have a specific grant number.

“AN received funding from the JST Global Science Campus ROOT Program.”

Thank you for your advice. We will prepare and implement our data sharing plan immediately after submitting the revised manuscript.

L241-245: We have added an ethics statement in the Materials and Methods section. According to the Regulations for Animal Experimentation and the Ethics Guidelines for Research Involving Human Subjects of Kobe University, ethical review was not required because the study involved only plants and invertebrate insects. Field sampling was conducted on private land owned by the first author’s family with permission.

Done. We have added captions for all Supporting Information files at the end of the manuscript and updated in-text citations accordingly.

Done. We have reviewed the recommended publications and cited those relevant to our study.

Additional Editor Comments:

Each of the reviewers has, in addition to the general comments and statements of concern or suggested changes, also offered additional specific comments in the article PDF which is attached to the authors in the submission platform. The authors must carefully access these, address each of the comments and, in resubmitting a thoroughly revised manuscript, include a separate document detailing how each of these issues has been addressed point by point, or a robust rebuttal as to why they disagree with suggested changes

The formatting of the document has not adhered to the guidelines of PLOS ONE. In assessing the revised manuscript, the strict compliance to formatting guidelines (structure, font sizes and styles as well as arrangement of the various component parts) will be the first criteria before the paper can progress any further, failure to which it will have to be rejected outright. For instance, in-text citation style uses numbers, not author-and-year; they must also be enclosed in square brackets, and listed in the corresponding number orders under References at the end.

Tables and their captions must be immediately below paragraphs they are first mentioned in;

Figure captions must be in the positions below paragraphs they are first mentioned in

Refer to submission guidelines for strict compliance will all requirements

We have also revised the manuscript to fully comply with PLOS ONE formatting guidelines, including citation style, reference numbering, and placement of tables and figures.

Reviewers' comments:

Reviewer #1: Dear Authors,

Thank you for submitting your manuscript to Plus One. The study addresses a valuable question about how fertilization shapes aphid performance and interspecific competition on pea plants. While your work has merit, the manuscript, in its current format, does not meet the journal’s standards for publication. Several important methodological and interpretive issues need to be addressed, and I have outlined them below with guidance to help you prepare a stronger revision. A separate document with detailed, itemized suggestions is attached.

Main issues to consider and revise:

1- Trait scope and depth:

- The current trait set (aphid counts and plant biomass) is too limited to robustly infer mechanisms of plant response to aphid infestation and to establish the influence of fertilization on competitive outcomes.

- Consider adding additional plant responses (e.g., chlorophyll content, photosynthetic rate, stomatal conductance, tissue nutrient concentrations, defensive metabolites) and aphid performance metrics (e.g., reproduction rate, development time, adult longevity, honeydew production) to better capture trophic interactions and tolerance mechanisms.

We agree that expanding plant and aphid trait measurements (e.g., chlorophyll, photosynthesis, tissue nutrients, defensive metabolites, reproduction and longevity metrics) would strengthen mechanistic inference. However, these variables were not measured in the present experiment, which was designed as a first step to isolate fertilization effects on population trajectories under controlled conditions.

L496-505: We now acknowledge this explicitly as a limitation and outline it as a priority for future work.

2- Proof of fertilization effect on species-specific aphid numbers:

Clarify how fertilization differentially affected the two aphid species. Provide explicit experimental design details and statistical tests that support the claim that changes in aphid counts by species are attributable to fertilizer treatment rather than other factors.

348-360; To avoid confounding by interspecific interactions, we analyzed only single-species treatments using two-way ANOVA (Species × Fertilizer) on AUC, peak, and end-point counts (log-transformed). Fertilization significantly reduced M. crassicauda performance but not A. craccivora. The Species × Fertilizer interaction was significant for end-point counts (F₍₁,₂₀₎ = 8.94, p = 0.0072), confirming species-specific effects. Methods and results have been added to the revised manuscript and Figure 2.

3- Containment and movement between pots:

Describe how you prevented aphid movement between pots (physical separation, barriers, spatial layout, or any measures to minimize cross-transfer). If movement was possible, discuss its potential impact on results and how it was mitigated or accounted for in analyses.

L185-190: We have added details on containment and spacing. Each pot was placed in an individual tray (24 × 17 × 5 cm) with a 2 cm water layer acting as a moat, and pots were spaced at least 10 cm apart (Fig 1c). Alate wings were removed upon detection. These measures effectively prevented aphid movement between pots.

4- Plant tolerance assessment

Biomass alone is insufficient to characterize plant tolerance to herbivory. Include additional indicators (growth rates, tissue allocation patterns, physiological stress indicators, biochemical markers) to support conclusions about tolerance under different nutrient regimes.

L496-505; We agree that biomass alone is insufficient to assess tolerance. As these measurements were not collected in the present study, we now acknowledge this as a limitation and note a plan to include tolerance indicators (growth rate, tissue allocation, physiological/biochemical stress markers) in future work.

5- Repetition and clarity regarding trait choices

There are repeated points about the limited trait set. Consider consolidating these into a single, clear rationale for expanding trait measurements and present a concrete plan for the expanded suite of traits.

L496-505; We consolidated all trait-related limitations and plans into one clear paragraph and removed repetition. No additional trait paragraphs were added.

6- Methods: measurement references

The Materials and Methods section should include explicit references or protocols for all measurement techniques. Provide enough detail for replication, including instrument models, measurement conditions, sampling timing, and data processing steps.

We added detailed methodological references rather than generic protocol citations. Specifically, we now (1) cite Moritsu (1982) for aphid identification, (2) reference Staley et al. (2011) for experimental design rationale, and (3) include key statistical references (Fox & Weisberg, 2019; Kaplan & Me

---

## [Editor Report · Decision Letter 1]

16 Nov 2025

Fertilization reduces aphid performance but does not alter competitive exclusion between specialist and generalist species

PONE-D-25-33799R1

Dear Dr. Saga,

We’re pleased to inform you that your manuscript has been judged scientifically suitable for publication and will be formally accepted for publication once it meets all outstanding technical requirements.

Kind regards,

Nickson Erick Otieno, PhD

Academic Editor

PLOS ONE

Additional Editor Comments (optional):

Thank you for adequately addressing most of the reviewers’ concerns and suggested changes in your revision.

The following few but important corrections need to be made before the document may oroceed to the next stage

In Table 2, what us the meaning of  ‘group2’ as a column header? Does that mean there is a ‘group1’ somewhere?Please ensure that column header/title first letter is capitalizedPlease limit decimal points (in tables and in-text) to no more than 4Ensure the manuscript title is font size 8Line 254: please define fully the acronym SD for standard deviation when it is first mentionedRegarding figure 4, you should provide better images for the two species of aphid – larger in size, higher resolution and better distinguished. Refer to submission guidelines and specifically how to use the freely available PACE tool for formatting figures to standards and quality  suitable for publication in PLOS ONEI strongly advice to replace the term ‘aphid performance’ with reference to aphids throughout the paper (including the title) with ‘intensity of herbivory’, or ‘plant consumption rate’ since its essence, ‘performance is very loosely vague and could many a broad range of things including not just feeding and reproduction but also growth, survival rate, health status and predator avoidance etc., all which you did not measure/quantify. Alternatively, define performance specifically but completely at end of introduction and in methods section to expressly demonstrate that it entailed only feeding and population growth,As a matter of fact, when  ‘performance if mentioned in the text, readers need to be able to distinguish whether you are referring to host plant performance or aphid herbivore performance, This is because from Line 130-132, you state that “***We quantified plant response at the whole-plant level (survival time and biomass change) as integrative indicators of host performance under herbivory, rather than physiological tolerance.***’ So here you are not talking about aphid performance anymore

The above are critical corrections required so, apart from effecting the changes in the t text itself, ensure that you also submit a separate account of how u=you have addressed them in your revised manuscript
---

## [Editor Report · Acceptance letter]

PONE-D-25-33799R1

PLOS ONE

Dear Dr. Saga,

I'm pleased to inform you that your manuscript has been deemed suitable for publication in PLOS ONE. Congratulations! Your manuscript is now being handed over to our production team.

Kind regards,

on behalf of

Dr. Nickson Erick Otieno

Academic Editor

PLOS ONE